# Cause of Color Modification in Tanzanite after Heat Treatment

**DOI:** 10.3390/molecules25163743

**Published:** 2020-08-16

**Authors:** Teerarat Pluthametwisute, Bhuwadol Wanthanachaisaeng, Chatree Saiyasombat, Chakkaphan Sutthirat

**Affiliations:** 1Department of Geology, Faculty of Science, Chulalongkorn University, Bangkok 10330, Thailand; chakkaphan.s@chula.ac.th; 2Gem and Jewelry Program, College of Creative Industry, Srinakharinwirot University, Bangkok 10110, Thailand; bhuwadol@g.swu.ac.th; 3Synchrotron Light Research Institute (Public Organization), Nakhon Ratchasima 30000, Thailand; chatree@slri.or.th

**Keywords:** tanzanite, heat treatment, zoisite, color

## Abstract

Natural tanzanites usually show strongly trichroic coloration from violet to blue, and brown colors in different directions. However, this characteristic is easily changed to violet-blue dichroism after heat treatment. Moreover, the cause of color modification after heating is still controversial. A few researchers have previously suggested that trace amounts of either vanadium or titanium substituted in aluminum site should be the main determinant of color after the heat treatment. Alteration of either V^3+^ to V^4+^ or Ti^3+^ to Ti^4+^ may relate to light absorption around 450–460 nm, which is the main cause. UV/vis/NIR absorption spectroscopy and X-ray absorption spectroscopy (XAS), a utility of synchrotron radiation, were applied for this experiment. As a result, the violet-blue absorption band (centered around 450–460 nm) as well as green absorption band (centered around 520 nm) were obviously decreased along the c-axis after heating, and XAS analysis indicated the increasing of the oxidation state of vanadium. This result was well supported by the chemical composition of samples. Consequently, vanadium was strongly suggested as the significant coloring agent in tanzanite after heat treatment.

## 1. Introduction

Tanzanite, a violet-blue vanadium-bearing variety of zoisite (Ca_2_Al_3_[Si_2_O_7_][SiO_4_]O(OH)) [1], has been long known and has become a highly demanded precious stone in the world gem market owing to its fantastic color appearance. Natural tanzanites usually show strong trichroism between violet, blue, and brown (or green) colors (Figure 1). Tanzanite was first discovered in 1967 [2] at Merelani area, north-eastern Tanzania (East Africa), by Jumanne Ngoma, a gypsum miner who was officially acknowledged by the Tanzanian government. The deposit only covers around 7 kilometers and its estimated life expectancy for mining is, unfortunately, less than 20 years [3]. Moreover, natural high-quality violet-blue colored tanzanites are rare; therefore, heat treatment has been applied to lower quality stones owing to the high demand in the market.

Zoisite, including tanzanite variety, is an orthorhombic sorosilicate mineral whose ideal formula can be defined as A_2_M_3_Si_3_O_12_OH [4], where A sites are generally substituted by divalent cations (e.g., Ca, Mn, Sr, Pb), while M sites are mainly occupied by trivalent cations (e.g., Al, Fe, Mn, Cr, and V) [5,6,7]. Its structure contains the base of edge sharing octahedral (M1,2) chain linked by double Si_2_O_7_ (T1, T2) and single SiO_4_ (T3) tetrahedra. This edge sharing octahedral chain in zoisite also has additional M3 octahedra attaching on one side of the chain [7] (Figure 2). The structure of zoisite is slightly different from clinozoisite (a member of epidote supergroup), with the additional M3 octahedra attaching on both sides along the b-axis of the octahedral M1 chain and the octahedral M2 chain being isolated from the M3-combined M1 chain (see also Figure 2). According to this information, the Commission on New Minerals, Nomenclature, and Classification of the International Mineralogical Association (IMA) has recently defined all members of epidote supergroup as monoclinic minerals, and thus tanzanite, a variety of orthorhombic sorosilicate zoisite, no longer belongs to the epidote supergroup [8,9,10].

Tanzanite has been heated to improve the color to sapphire-like blue color, which is highly demanded in the market [2,11]. It is believed that the violet-blue color in the tanzanites is owing to the presence of vanadium, which replaces aluminum in the octahedral site [3,5,12,13,14,15,16,17]. Heating tanzanite at approximately ~500 °C usually produces the disappearance of the yellow color in that particular direction in which the absorption around 450–460 nm is decreased, and alteration of pleochroism from trichroic to dichroic violet to blue color [3,13,14,17]. Heating tanzanite at too low or too high a temperature may lead to never reaching the desirable color alteration; no color alteration is observed below 250 °C [13], whereas color appears to be faded at a high temperature [3]. In situ heating spectroscope was used to reveal the optimum temperature (550 °C), which causes the disappearance of the absorption around 450–460 nm [3]. Color alteration behavior after heat treatment can also be observed in other gemstones. For example, blue intensification and removal of green tint in blue-colored variety (aquamarine) of beryls were performed by heating at 300–700 °C, leading to the reduction of Fe^3+^ to Fe^2+^. Consequently, Fe^3+^ absorption bands at 375 and 425 nm are decreased, while Fe^2+^ band at 810 nm is significantly intensified. However, heating aquamarines at higher temperature (e.g., 1100 °C) may lead to an opaque appearance, which relates to the disappearance of the 810 nm absorption band [18]. Another example is in diaspore (zultanite), where the chromophores involved in the variation in the color are owing to the presence of Ti-Fe and are associated with the intervalence-charge-transfer mechanism [19].

However, coloration in tanzanite after heat treatment has been still a questionable subject for decades. Previous researchers speculated that coloration of heated tanzanite is owing to oxidation changing of either vanadium or titanium. Vanadium may have been oxidized from V^3+^ to V^4+^ after heat treatment [5,14]; however, Olivier [3] argued that the coloration of tanzanite after heat treatment is owing to an oxidizing of titanium from Ti^3+^ to Ti^4+^. Moreover, a more recent inconsistency by Smith [16] quoted that the change in color of brown tanzanite after heating is attributed to a conversion of V^4+^ to V^3+^, as well as Bocchio et al. [17], who also introduced that the V/Ti ratio should play an important role in the coloration of tanzanite without any concern about oxidation states of both elements in natural and heated tanzanites. Since then, the controversial topic of oxidation states of the color-causing element in tanzanite has still not been settled.

X-ray absorption spectroscopy (XAS), a synchrotron radiation utility, is a potential atomic probe technique that is sensitive to the oxidation state and local structure of absorbing element. XAS is basically divided into two regions: X-ray absorption near edge structure (XANES) and extended X-ray absorption fine structure (EXAFS). XANES is a useful spectroscopic technique to determine the oxidation state and structural symmetry, while EXAFS is used to explore the local structure of the absorbing atom including the distance of neighboring atoms, coordination number, and disorder of neighboring atoms. A probed element with various oxidation states and local structure will affect the shift of the absorption edge (edge energy, E0) [20,21,22,23] and characteristic XANES features. Therefore, this study is attempted to ascertain the main cause of modified coloration in tanzanite after heat treatment. Changing in oxidation states of vanadium and titanium is the focus in this study.

## 2. Results

### 2.1. General Properties

The physical properties of all tanzanite samples are summarized in Table 1. They ranged from 0.565 to 1.26 carats in weight with specific gravity (S.G.) of 3.32–3.38. Refractive indices (R.I.) and birefringence fell within the ranges of 1.691–1.701 and 0.08–0.09, respectively. All samples were inert under UV lamp. Their natural trichroic colors and dichroic colors after heating observed along different directions are presented in Figure 3.

### 2.2. Chemical Compositions

Major and trace compositions, resulting from EPMA analyses, are presented in Table 2. A.P.F.U. were recalculated on the basis of 13 oxygen atoms. Three analytical spots were carried out on the C-axis (yellowish color). SiO_2_, Al_2_O_3_, and CaO ranged from about 39.0 to 40.5 wt%, 33.0 to 34.0 wt%, and 24.0 to 24.5 wt%, respectively. V_2_O_3_ content yielded <0.5 wt%, whereas TiO_2_ content was even lower than 0.05 wt%, and other trace elements including MgO, Cr_2_O_3_, FeO, MnO, Na_2_O, and K_2_O were negligible and mostly lower than 0.1 wt%. Apart from the major contents of Si, Al, and Ca, the EPMA data also indicated that V is the most significant trace chromophore in these samples.

### 2.3. Heat Treatment

As expected, heating with optimum temperature (550 °C) dramatically effected the colors of tanzanite in all directions (Figure 3). Yellow tints in the C-axis were obviously removed after the heating experiment. Figure 3 also presents the color codes of natural colors and colors after heat treatment along the three main directions; for instance, the sample T04 yielded color alteration from yellowish orange (yO 6/3) to violet (V7/4) along the c-axis, green-blue (GB 6/1) to bluish violet (bV 6/5) along the b-axis, and bluish purple (bP 6/3) to violet (V7/4) along the a-axis. In general, their trichroic colors appeared to be changed to dichroic colors after heat treatment. The C-axis direction always shows the strongest color modification compared with the other direction.

### 2.4. UV/VIS/NIR Spectroscopy

UV/vis/NIR spectroscopy revealed a representative spectra of natural and heated tanzanite sample (Figure 4). This natural tanzanite sample T04 showed the highest absorption band with a peak at about 460 nm (violet-blue range), particularly in the c-axis, whereas the absorption band with a peak at around 585–600 nm (yellow range) appeared to be the most typical in all directions. In fact, the absorption band with a peak at 460 nm also presented slightly in the a- and b-axes, but this absorption was significantly decreased in all directions after heating. On the other hand, the absorption band (peak around 585–600 nm) still remained as shown in Figure 4.

In summary, decreasing the absorption band with a peak at 460 nm (violet to blue range) was obviously observed in all directions after heating experiment (Figure 4), whereas the absorption band with a peak around 585–600 nm was unchanged. Only in c-axis, was the absorption band with a peak at 585 nm slightly increased after heating. It should be notified that natural tanzanite usually shows an important absorption band with a peak around 460 nm, which is possibly caused by either vanadium or titanium, as suggested by previous researchers [3,5,12,13,14,15,16,17].

### 2.5. X-Ray Absorption Spectroscopy (XAS)

Edge energy (E0) data in the natural tanzanite were increased to higher energy of vanadium after heat treatment, as summarized in Table 3. For instance, edge energy of natural tanzanite sample T06 was 5478.95 eV and subsequently reached up to 5479.15 eV. On the other hand, titanium in the same sample appeared to be decreased in edge energy from 4981.00 eV in natural tanzanite to 4980.70 eV after heating experiment (Table 3). All analytical data of E0 obtained from all samples are compared in the graphic presentation in Figure 5 and Figure 6 for V and Ti, respectively. Most samples obviously showed increasing edge energy of vanadium and decreasing edge energy of titanium. The change in edge energy refers to changing of the oxidation state; the higher energy indicates the higher oxidation state [20]. Therefore, V tended to oxidize after heating at 550 °C in atmospheric environment, whereas Ti behaved in the other way.

## 3. Discussion

V and Ti have previously been proposed to be the main chromophore allocated within Al-octahedral site of tanzanite; both elements could have changed oxidation states after heating [3,5,7,12,13,14]. V^3+^ should be oxidized to V^4+^ after heating and cause a sapphire-like blue color [2,5,11,14]. However, Olivier [3] has later suggested that heating could oxidize Ti rather than vanadium in tanzanite. He has also proposed that Ti^3+^ is responsible for the red/green/blue color in the natural tanzanite and appears to be oxidized to colorless Ti^4+^ after heating.

On the basis of the results of this investigation, natural tanzanite showed a mainly purplish blue, green-blue to green-yellow, and yellow to yellowish orange in the a-, b-, and c-axes, respectively. After the heating experiment, the yellow to yellowish orange color along the c-axis turned into violet and the bluish purple changed to violet along the a-axis, whereas the yellowish/greenish blue was altered to a blue-dominated color in the b-axis. Consequently, its pleochroism changed from trichroic to dichroic coloration, accordingly. Therefore, green and/or yellow colors were conclusively removed by heating at 550 °C (see Figure 3). The most significant change can be observed along the c-axis, which contained the most intense yellow component in the natural stone.

In general, different absorption spectra of natural tanzanite samples appear in each particular direction (a-, b-, or c-axis), for example, in sample T04 (Figure 4). A strong absorption band in the yellow region (peak at 585–600 nm) can be recognized in all directions. This absorption band should involve violet coloration, which can be explained using the role of complementary colors suggested by Nassau [24]. The visible color spectrum arranged in the triangular presents pairs of complementary colors on the opposite side such as blue-orange, green-red, and yellow-violet. Each pair of complementary colors provides a white color or no color. Therefore, selective absorptions in the visible spectrum may yield the combination of colors of which their complementary pairs are absorbed [24]. The main absorption band with a peak at about 585–600 nm (yellow range) still remains after heating; therefore, it should be the main cause of violet coloration, particularly in heated tanzanite. For instance, violet (V7/4 in sample T04) appears along a- and c-axes, which is clearly caused by such an absorption band (see Figure 4a,c). On the other hand, the peak of this absorption band slightly shifts towards the orange range (600 nm) in the b-axis (see Figure 4b), which leads to bluish violet (bV5/5) being present, based on the same principle.

Regarding colors in natural samples, the yellowish orange (yO 6/4) color of natural tanzanite sample T04 in the c-axis is clearly effected by very strong absorption band peaked around 460 nm covering the blue region (Figure 4c), which this absorption pattern decreases significantly in both the a- and b-axes, yielding bluish purple (bP 6/3) and green-blue (GB 6/1), respectively. This absorption band is clearly an indication of natural tanzanite (unheated), as also suggested by previous workers [3,5,12,13,14,15,16,17].

Consequently, it is implicit that the trichroic bluish purple, green-blue to green-yellow, and yellow to yellowish orange colors of the a-, b-, and c-axes of natural tanzanite are obviously influenced by different intensities of absorption band (peaked around 460 nm) in the blue region, as explained above. This absorption band is absent in all directions after heating at 550 °C, which leads to dichroic violet (a- and c-axes) and blue (b-axis). In addition, a more stable absorption band in the yellow region (peak at 585–600 nm) presented in all directions of both natural and heated tanzanite should be the main color causing blue-violet. A slight shifting of the peak of this absorption band from 585 to 600 nm may induce different shades such as purplish to bluish, accordingly.

XAS analyses give rise to new evidence indicating that the edge energy of vanadium in most samples changes to the higher oxidation state, whereas titanium yields lower edge energy and decreases the oxidation state after heat treatment (Table 3, Figure 5 and Figure 6). It has been suggested by Olivier [3] that Ti^3+^ causing red/green/blue color in natural tanzanite should be oxidized to Ti^4+^, then contributing no color. However, the results obtained from this study indicate that the oxidation state of titanium is likely decreased after heat treatment. In fact, this study reveals the increasing of vanadium edge energy in most samples, which clearly relates to more intense violet and blue coloration, which is a result of extreme decreasing of absorption in the blue region.

EPMA analyses indicate that all tanzanite samples are similar in composition and close to the idealized formula Ca_2_Al_3_[Si_2_O_7_][SiO_4_]O(OH). Their major compositions are composed, on the basis of 13 oxygen atoms, of Si (3.1–3.2 apfu), Al (3.1 apfu), and Ca (2.0–2.1 apfu). Moreover, they also confirm that vanadium contents (up to 0.027 apfu) are much higher than titanium (≤0.002 apfu) in these tanzanite samples (see Table 2). Electron microprobe analyses of tanzanite in this present work represent the average results of 0.29 wt% V_2_O_3_, 0.01 wt% TiO_2_, 0.03 wt% Cr_2_O_3_, and 0.01 wt% FeO, which are compatible with those recently reported by Bocchio et al. [17], including about 0.39 wt% V_2_O_3_, 0.01 wt% TiO_2_, 0.07 wt% Cr_2_O_3_, and 0.00 wt% FeO, as well as those of Barot and Boehm [15], who earlier presented averages of 0.15 wt% V_2_O_3_, 0.03 wt% TiO_2_, 0.08 wt% Cr_2_O_3_, and 0.01 wt% Fe_2_O_3_. However, it is in contrast to those reported by Olivier [3], who presented the lower average V_2_O_3_ contents of 0.06 wt%, but comparable averages of 0.03 wt% TiO_2_, 0.04 wt% Cr_2_O_3_, and 0.02 wt% FeO.

V/Ti ratios obtained from this work show that V contents are about 13 to 40 times greater than Ti content, which fits well with the color-causing model of tanzanites. V content was quoted to be 60 times greater than Ti content in blue colored samples, but decreasing to 16 to 5 times was observed in yellowish brown and greenish yellow colored zoisites [3,13,17]. Moreover, the V_2_O_3_/Cr_2_O_3_ ratios of all tanzanites in this work are greater than two, which typically yields a blue color, whereas lower than two is usually detected in green tanzanite [25]. Therefore, it is clear that the main chromophore in these samples is vanadium, which appears to have contributed to violet and blue coloration after heat treatment.

## 4. Materials and Methods

Five rough tanzanite samples were collected based on their strong trichroism, which clearly showed various colors in different directions. During rotation of the sample, color changing was clearly observed such as yellowish orange to orangey yellow, greenish yellow to greenish blue to bluish green, and bluish purple along particularly directions, which were marked. Subsequently, samples were then cut and polished following those marked directions and their colors were compared with the GIA standard color GemSet. Colors along these particular directions were also observed again after heating experiment. In addition, absorption spectra were analyzed using a PerkinElmer-LAMBDA 900 UV/VIS/NIR spectrophotometer based at Department of Earth Sciences, Kasetsart University (PerkinElmer LAMBDA, Waltham, MA, USA); the operating conditions were set with 2.00 slit size and recording range between 200 nm and 1500 nm. Chemical compositions of the samples were analyzed using an electron probe microanalyzer (EPMA), JEOL model JXA 8100 (Musashino, Akishima-shi, Tokyo, Japan), while the heating experiment was carried out using a Linn-HT-1800-Vac high temperature furnace (Linn High Therm, Frankenhausen, Germany). Both EPMA and high temperature furnace were facilitated by Department of Geology, Chulalongkorn University. Analytical conditions of EPMA were set at 15 kV accelerating voltage and about 2.5 × 10^-8^ A probe current with focus electron beam (<1 µm). Mineral standards and some synthetic oxide standards were selected appropriately for calibration including jadeite (NaAlSi_2_O_5_) for Na, fayalite (Fe_2_SiO_4_) for Fe, wallastonite (CaSiO_3_) and potassium titanium phosonate (KTiPO_4_) for Ti and K, synthetic corundum (Al_2_O_3_) for Al, synthetic periclase (MgO) for Mg, synthetic quartz (SiO_2_) for Si, synthetic manganesite (MnO) for Mn, synthetic eskolaite (Cr_2_O_3_) for Cr, and lead vanadium germanium oxide (PbV Geoxide) for V. The heating experiment was carried out in atmospheric condition at the maximum temperature of 550 °C with approximately a 1.8 °C/min heating rate without holding time prior to cooling down naturally in the furnace. XAS was investigated at Beamline 1.1W: Multiple X-ray Techniques (MXT) experimental station (Nakhon Ratchasima, Thailand) at the Synchrotron Light Research Institute (Public Organization) in Thailand (Figure 7). This beamline has a 2.2 Tesla multipole wiggle as the X-ray source and equipped with Si (111) double monochromator. A 19-element Ge detector was used as a florescence detector to detect probed element with low concentration. All XAS analyses were examined using 1.65 (V) × 2.6 (H) mm beam size along the C-axis in all samples with an average of three scans to improve signal to noise ratio. A standard foil for each element was used to calibrate X-ray energies by setting the absorption edge energy of V and Ti to 5465 eV and 4966 eV, respectively. An energy scan ranging from 200 eV below to 200 eV above the absorption edge energy was used with the highest collection time of 2 s and 7 s per point for vanadium and titanium, respectively. Furthermore, V_2_O_3,_ V_2_O_4_, Ti_2_O_3_, and TiO_2_ were measured as chemical standards in transmission mode using ionization chambers. However, these standards are not the same mineral group as tanzanite, and thus are used as a reference point for relative change, but are unable to inform the exact oxidation state of V and Ti in the samples. The absorption edges of V and Ti from XANES spectra were then extracted from their first derivative graph and presented in a calibration curve.

## 5. Conclusions

Physical and chemical properties of the studied tanzanites were observed and measured using basic and advanced gemological instruments in combination with X-ray absorption spectroscopy (XAS). These results can be used to demonstrate the coloration in natural and heat-treated tanzanites, which can be recapitulated as follows.

(I) More desired violet-blue color in tanzanite can be improved by thermal process; (II) the absorption band with a peak around 460 nm (blue range) is the main cause of color in different directions (trichroism) of natural tanzanite, whereas the absorption band in the yellow region (peak at 585–600 nm) similarly presents in all directions and still remains after heating; (III) after heating, the c-axis shows the most significant change in color compared with the other directions—it clearly relates to the disappearance of the 460 nm absorption band; and (IV) XAS spectra show an increasing of edge energy of vanadium, which indicates the higher oxidation state after heat treatment—this should cause the disappearance of the 460 nm absorption band and induce more intensive blue-violet color in heated tanzanites.

## Figures and Tables

**Figure 1 molecules-25-03743-f001:**
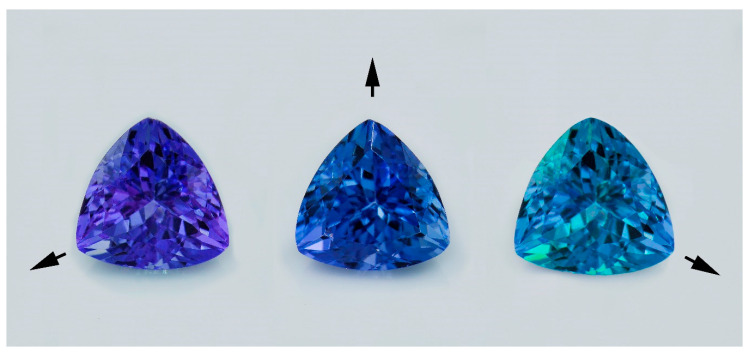
Trichroism of a natural tanzanite rotated in three directions showing violet, blue, and green pleochroic colors (photo by T. Sripoonjan, GIT).

**Figure 2 molecules-25-03743-f002:**
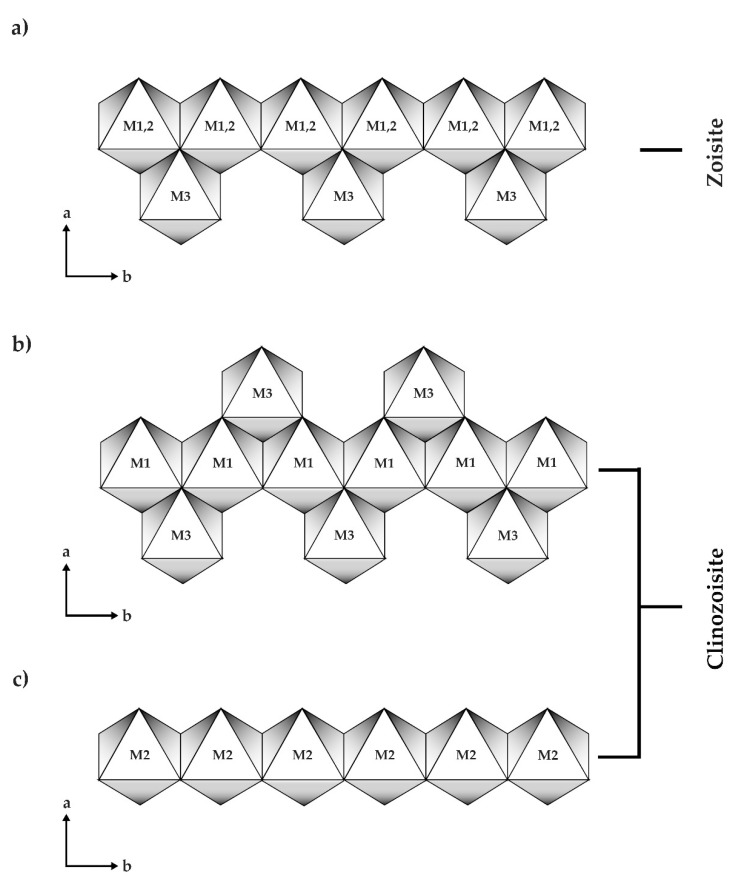
(**a**) Zoisite octahedral chain (M1, 2) combined with additional octahedra (M3) on only one side along b-direction, whereas (**b**) additional octahedra (M3) located on both sides of clinozoisite octahedral chain (M1) with (**c**) isolated octahedral chain (M2) (modified after Franz and Liebscher [7]).

**Figure 3 molecules-25-03743-f003:**
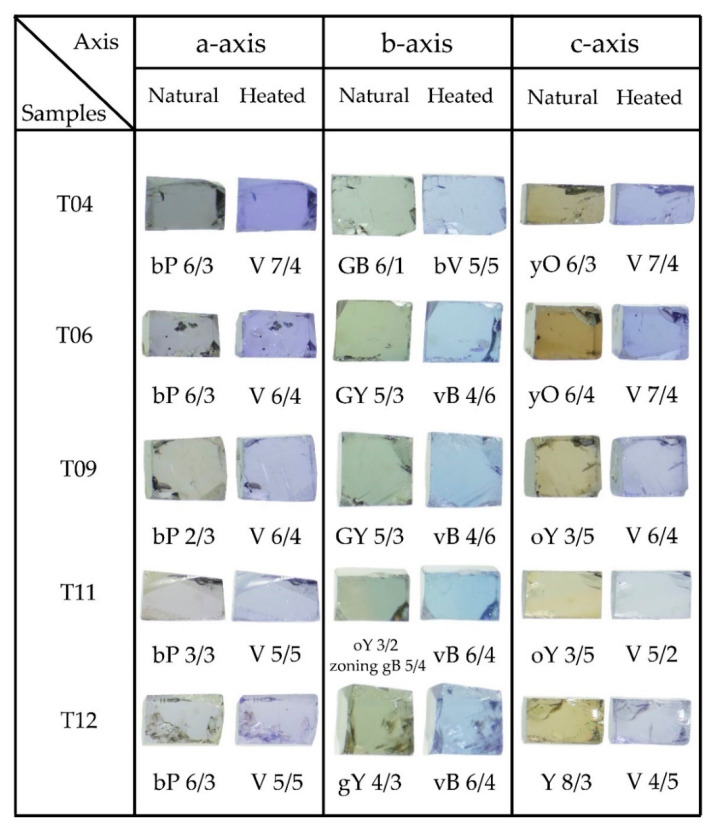
All tanzanite samples show natural trichroic colors including purple, green, and yellow dominating along the directions of a-, b-, and c-axes, respectively, as suggested by [3,13,14,17]; they turned to dichroic coloration between violet color (along the a- and c-axes) and blue color (along the b-axis) after the heating experiment.

**Figure 4 molecules-25-03743-f004:**
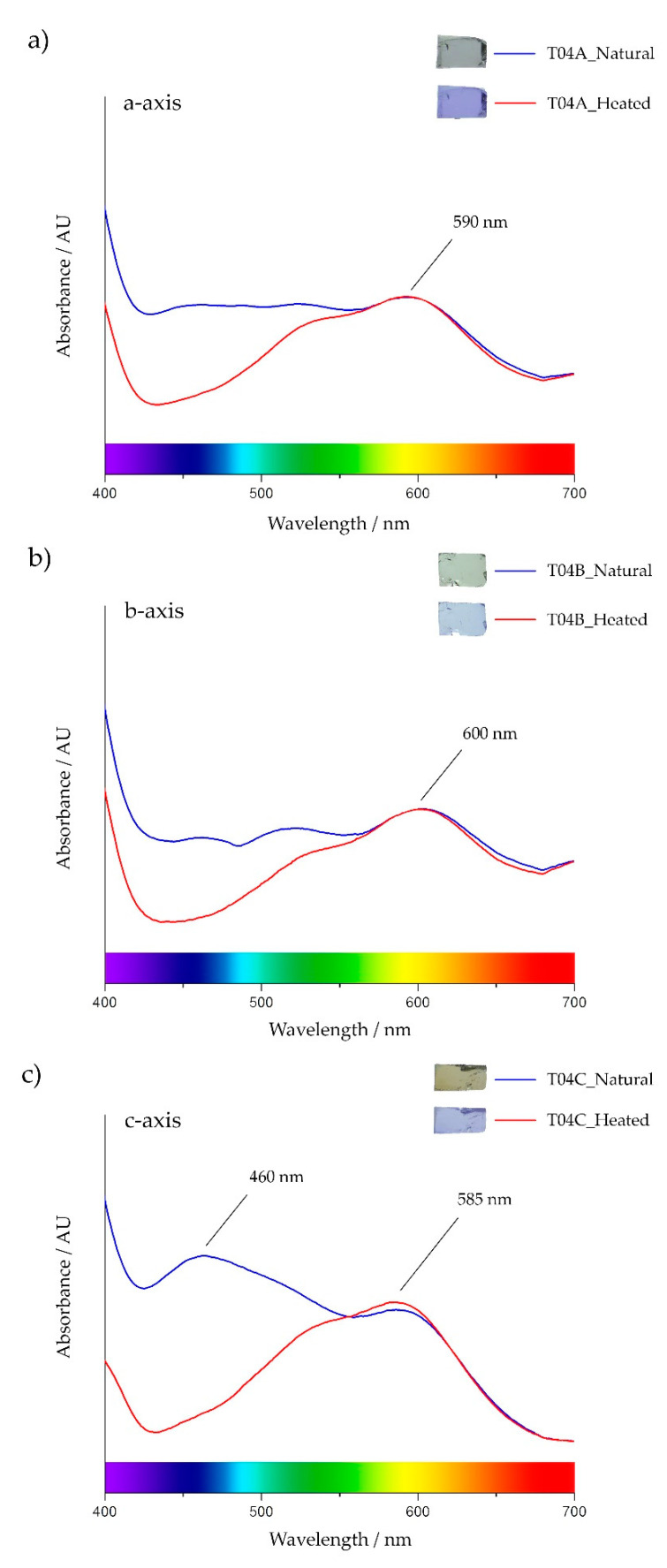
Representative absorption spectra of natural (blue spectrum) and heated (red spectrum) along the a-axis (**a**), b-axis (**b**), and c-axis (**c**) of a tanzanite sample (T04) (visible spectrum modified after Nassau [24]).

**Figure 5 molecules-25-03743-f005:**
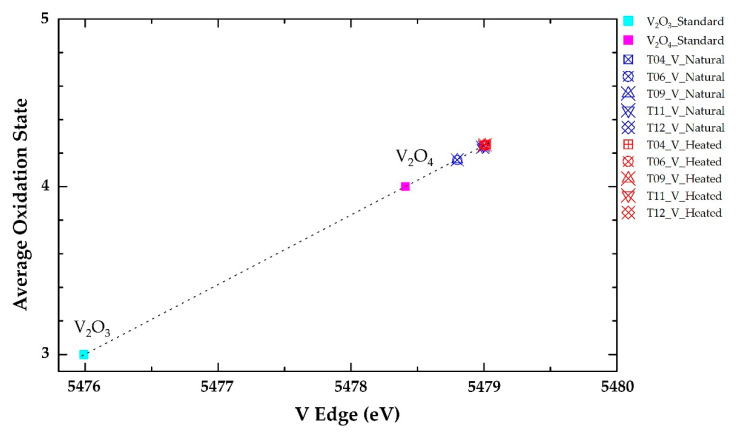
Plots of absorption edge energy (E0) of vanadium in natural tanzanite samples and after heating experiment with a calibration curve (dotted line) fitted by standard V_2_O_3_ and V_2_O_4_; oxidation states of vanadium in most samples appear to increase after heating.

**Figure 6 molecules-25-03743-f006:**
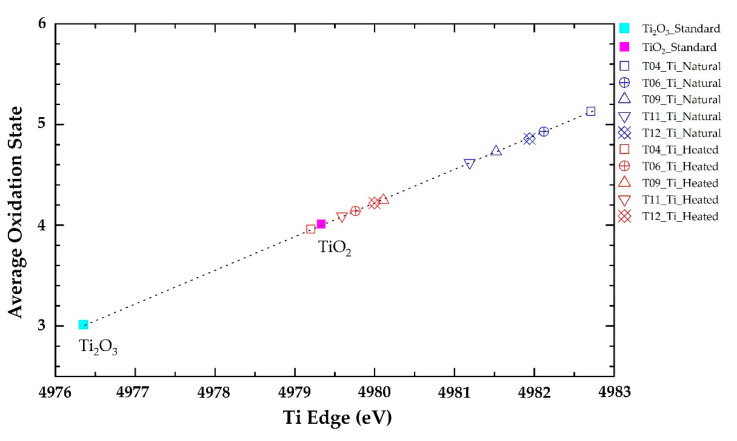
Plots of absorption edge energy (E0) of titanium in natural tanzanite samples and after heating experiment with a calibration curve (dotted line) fitted by standard Ti_2_O_3_ and TiO_2_; oxidation states of titanium tend to decrease after heating.

**Figure 7 molecules-25-03743-f007:**
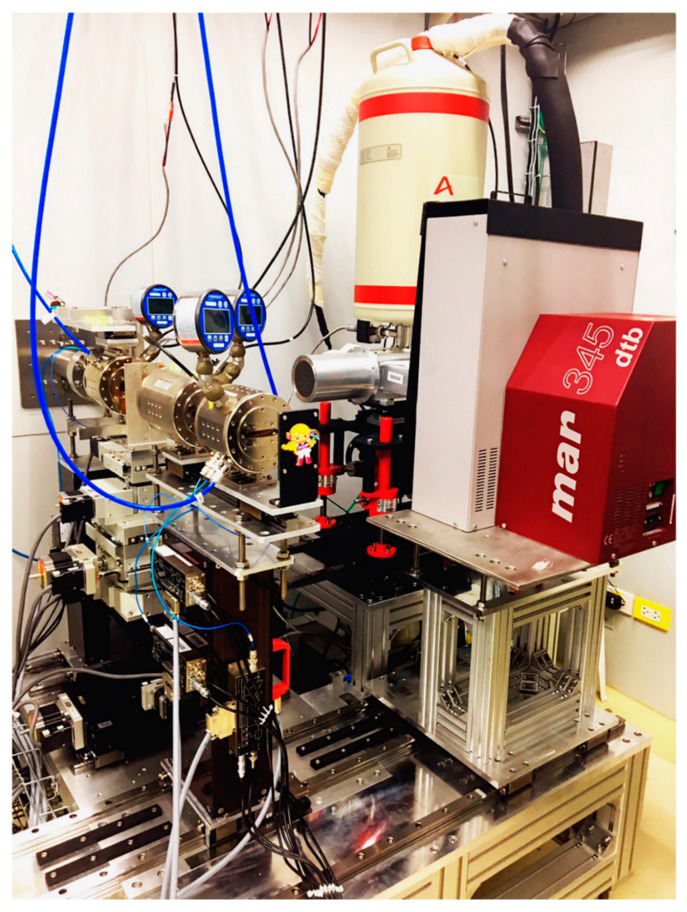
Beamline 1.1W: Multiple X-ray Technique station (photo by C. Saiyasombat).

**Table 1 molecules-25-03743-t001:** General properties of tanzanite samples in this study.

Sample/Properties	T04	T06	T09	T11	T12
Refractive Indices	1.692–1.700	1.691–1.700	1.691–1.700	1.693–1.701	1.692–1.701
Birefringence	0.008	0.009	0.009	0.008	0.009
Specific Gravity	3.33	3.35	3.37	3.32	3.38
UV Fluorescence	Inert	Inert	Inert	Inert	inert

**Table 2 molecules-25-03743-t002:** Electron probe microanalyzer (EPMA) analyses of major and trace compositions of tanzanite samples under this investigation.

Sample/Oxides (wt%)	T04_1	T04_2	T04_3	T06_1	T06_2	T06_3	T09_1	T09_2	T09_3	T11_1	T11_2	T11_3	T12_1	T12_2	T12_3
SiO_2_	39.78	39.04	39.04	40.02	40.21	39.79	39.86	39.42	39.50	40.12	40.49	39.43	39.46	39.77	39.25
TiO_2_	0.00	0.03	0.00	0.03	0.00	0.00	0.05	0.00	0.00	0.00	0.02	0.00	0.00	0.00	0.03
Al_2_O_3_	33.48	33.02	33.47	33.19	33.21	33.99	33.21	33.72	33.83	33.23	33.17	33.09	33.42	33.26	33.25
V_2_O_3_	0.39	0.43	0.38	0.35	0.38	0.39	0.29	0.20	0.29	0.25	0.25	0.25	0.18	0.06	0.21
Cr_2_O_3_	0.00	0.07	0.08	0.05	0.02	0.08	0.00	0.07	0.00	0.02	0.00	0.05	0.00	0.00	0.00
FeO	0.00	0.00	0.00	0.03	0.00	0.01	0.00	0.00	0.00	0.06	0.00	0.03	0.03	0.00	0.04
MnO	0.00	0.01	0.00	0.00	0.02	0.00	0.06	0.03	0.03	0.02	0.00	0.00	0.01	0.01	0.01
MgO	0.05	0.04	0.03	0.05	0.03	0.05	0.04	0.05	0.05	0.02	0.01	0.01	0.05	0.05	0.04
CaO	24.26	24.28	24.12	24.38	24.56	24.45	24.64	24.34	24.34	24.47	24.43	24.37	24.33	24.35	24.55
Na_2_O	0.00	0.01	0.00	0.00	0.01	0.00	0.00	0.00	0.02	0.00	0.02	0.00	0.01	0.01	0.00
K_2_O	0.01	0.00	0.01	0.01	0.00	0.00	0.01	0.01	0.00	0.00	0.01	0.02	0.01	0.01	0.00
Total	97.97	96.94	97.13	98.11	98.44	98.76	98.15	97.84	98.06	98.19	98.41	97.94	97.48	97.52	97.38
Atom Per Formula Unit based on 13 oxygens															
Si	3.128	3.110	3.101	3.144	3.149	3.107	3.134	3.107	3.105	3.149	3.168	3.128	3.120	3.141	3.112
Ti	0.000	0.002	0.000	0.002	0.000	0.000	0.003	0.000	0.000	0.000	0.001	0.000	0.000	0.000	0.002
Al	3.104	3.100	3.132	3.073	3.065	3.127	3.077	3.132	3.135	3.074	3.059	3.093	3.115	3.096	3.107
V	0.025	0.027	0.024	0.022	0.024	0.024	0.018	0.013	0.018	0.016	0.016	0.016	0.011	0.004	0.013
Cr	0.000	0.005	0.005	0.003	0.001	0.005	0.000	0.005	0.000	0.001	0.000	0.003	0.000	0.000	0.000
Fe	0.000	0.000	0.000	0.002	0.000	0.001	0.000	0.000	0.000	0.004	0.000	0.002	0.002	0.000	0.002
Mn	0.000	0.001	0.000	0.000	0.002	0.000	0.004	0.002	0.002	0.001	0.000	0.000	0.001	0.001	0.001
Mg	0.005	0.005	0.003	0.006	0.004	0.006	0.004	0.006	0.006	0.002	0.001	0.001	0.006	0.006	0.004
Ca	2.045	2.072	2.053	2.052	2.061	2.045	2.075	2.055	2.050	2.058	2.048	2.071	2.061	2.060	2.086
Na	0.000	0.002	0.000	0.000	0.002	0.000	0.000	0.000	0.003	0.000	0.003	0.001	0.001	0.001	0.000
K	0.001	0.000	0.001	0.001	0.000	0.000	0.001	0.001	0.000	0.000	0.001	0.002	0.001	0.001	0.000
Total	8.308	8.323	8.319	8.305	8.307	8.315	8.316	8.319	8.320	8.306	8.296	8.317	8.317	8.310	8.327

**Table 3 molecules-25-03743-t003:** Edge energy values of V and Ti obtained from K-edge X-ray absorption near edge structure (XANES) spectra of tanzanite samples before and after the heating experiment.

Samples	Edge Energy of V (eV)		Edge Energy of Ti (eV)	
	Before	After	Before	After
T04	5479.00	5479.02	4981.92	4979.20
T06	5478.95	5479.15	4981.00	4980.70
T09	5478.82	5478.97	4981.40	4980.90
T11	5478.75	5479.02	4980.83	4979.30
T12	5478.80	5478.95	4980.73	4980.63

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
