# Peer review of "Cause of Color Modification in Tanzanite after Heat Treatment"

_molecules, 2020, doi:10.3390/molecules25163743_

Round 1

Reviewer 1 Report

I think this work is interesting but can be accepted only after major revision. One of the main problems is the English form which is not adequate. I am not
a native speaker, I have proposed some corrections but I believe that English
should be thoroughly reviewed by a language specialist.

The work is aimed at understanding whether the cause of the color change in tanzanite is due to the oxidation of vanadium, titanium or both. However, the chemical analyzes reported in the paper show that, in these tanzanites, titanium is absent or present in very low quantities. Consequently, I believe that Ti cannot, in any case, be a chromophore agent and the change in color can only be attributed to vanadium. I also observe that Fe which is an important cromophore is very low but its contents are not different from those of Ti. All these points should be well explained in the text. In addition, data on the chemistry of other Tanzanites in literature should also be reported, especially with regard to chromophores, for comparison with the presented data.

In general I think that data on chemical composition of cromophores should include also analyses on trace elements by LA-ICP-MS. The analyses reported in this paper have been carried out only with EMPA that can be used only for major element analyses. The trace chromophore elements are very important for gem-quality minerals.

Move the paragraph of materials and methods after the introduction. Currently this paragraph is incomprehensibly at the bottom of the work.

 Discussion: The discussion is mostly based on a comparison with tanzanites in literature. You should have to add the values reported in literature. Also the English form of the whole chapter should be revised. The discussion between the lines 206 and 2012 is not supported by an adequate Ti contents.

The paragraph of the conclusions is too brief and the discussion should be deepened in more detail.

I report some corrections on English form:

 Lines 34-35: This sentence is not very clear. “a gypsum miner which officially acknowledged by the Tanzanian government”

Fig. 1: The caption is not very clear. Are the 3 photos related to the same gem oriented in 3 different ways or are they 3 different gems? This point should be clarified better.

The sentence between 42 and 45 is not well written and very long. The word “which” is not correct and the colon and semicolon in the last part should be deleted . The sentence should be written as follows: “where A sites are generally substituted by divalent 44 cations (e.g., Ca, Mn, Sr, Pb) while M sites are mainly occupied by trivalent cations (e.g., Al, Fe, Mn, Cr…”

Line 48: which is not correct. Its better to write “with”.

Line 61: I would write “ it is believed that the violet-blue color in the tanzanites is due to the presence of vanadium which replaces aluminum in the octahedral site…”

The sentence between 62 and 65 is not well written. I would write “usually produces the disappearance of the yellow color in that particular direction in which the absorption …”

Line 66: Please write “no color alteration is observed below 250..”

Line 67: “In-situ heating spectroscope was used to reveal the optimum temperature 68 (550oC) which causes the disappearance of the absorption around 450-460 nm…”

Line 73: please change the word “conflict” which is not appropriate.

Line 80: “that is sensitive..”

Liner 83: “EXAFS is used to define the environment” What is the meaning of environment? Not clear in this context.

Line 104: “were presented in Table 2. A.P.F.U. were recalculated on the basis of 13 oxygen atoms…”

Line 116: Yellow tints in the C-axis were obviously removed ..

The sentence between 224 and 225 is not clear.

The sentence of line 257 is badly written.

Caption of Fig. 3: Why "presumably"?

Caption of Fig. 6 . In the Figure 6 I see that Ti in the natural standard samples passes from oxidation number 4 to 3 but there are some plots that are near oxidation number 5..Can you explain it better? I did not know Ti could have oxidation number 5. Maybe add some references.

Caption of Fig.8: The description of the photo on my opinion is not meaningful: the photo only shows the equipment.  

Description of Tables 3 and 4: the Edge energy values for V and Ti reported in the two tables are very different. why? 

Reviewer 2 Report

PAPER TITLE: Cause of Color Modification in Tanzanite after Heat Treatment

AUTHORS: Pluthametwisute et al.

REFERENCE: Molecules-880165

REFEREE'S COMMENTS TO AUTHORS

This paper describes interesting results about the identification of the chromophores responsible of the colour of both raw and heated tanzanite (Ca2Al3(Si2O7)(SiO4)O(OH)) up to 550oC. The authors do a sensible interpretation about the modification in colour centres of this gemstone attributed to the redox processes involving V and Ti cations in tanzanite specimens studied by means of EPMA, XAS, XANES... It is known that the effect of thermal treatment can induce changes in the colour of some other quality-gem minerals, for instance in diaspore (zultanite), where the chromophores involved in the variation in the colour is due to the presence of Ti-Fe and associated with intervalence-charge-transfer mechanism (Garcia-Guinea et al., Effects of preheating on diaspore: Modifications in colour centres, structure and light emission. J. Phys. Chem Solids. 2005, 66, 1220-1227. DOI: 10.1016/j.jpcs.2005.04.001). To the best of my knowledge, the information of the manuscript shows very interesting and unpublished results never reported before. It is concise, well-written and the results are well-supported by figures, table and references. The paper falls within the scope of Molecules. I would recommend for publication after minor amendments.

Special comments:

- Introduction section.

I should add a paragraph with the corresponding references, in this section, describing a similar behaviour previously observed in some other minerals that are also linked to redox reactions induced by thermal treatment (for instance in diaspore) or ionizing or UV radiation (quartz).

- Materials and methods section.

Fig 5 and 6 might be improved

Fig 8 could be removed

- Reference section.

Modify this section accordingly

Round 2

Reviewer 1 Report

Dear Authors, the new version of your paper fully satisfies all my questions and  criticisms. So I agree. Thank you very much.